# Occurrence and Molecular Characterization of *Cryptosporidium* Infection in HIV/Aids Patients in Algeria

**DOI:** 10.3390/v15020362

**Published:** 2023-01-27

**Authors:** Malika Semmani, Damien Costa, Nassima Achour, Meriem Cherchar, Hanifa Ziane, Abdelmounaim Mouhajir, Venceslas Villier, Haiet Adjmi Hamoudi, Loic Favennec, Romy Razakandrainibe

**Affiliations:** 1Unité de Parasitologie-Mycologie El Hadi Flici Ex.El Kettar Hospital, Alger 16000, Algeria; 2EA 7510 ESCAPE, Université de Rouen Normandie, 76000 Rouen, France; 3Centre National de Référence-Laboratoire Expert des Cryptosporidioses, Institut de Biologie Clinique, Centre Hospitalo-Universitaire C. Nicolle de Rouen, 76000 Rouen, France; 4Service d’Infectiologie B, EHS El Hadi Flici Ex.El Kettar Hospital, Alger 16000, Algeria; 5Service de Parasitologie-Mycologie Hopital Central De l’Armée, Alger 16000, Algeria

**Keywords:** *Cryptosporidium*, HIV/AIDS patients, molecular epidemiology, Algeria, MENA region

## Abstract

The estimated prevalence rate of adults living with HIV infection in MENA is one of the lowest in the world. To date, no data on the genetic characteristics of *Cryptosporidium* isolates from HIV/AIDS patients in Algeria were available. This study aimed to identify *Cryptosporidium* species and subtype families prevalent in Algerian HIV-infected patients and contribute to the molecular epidemiology mapping of *Cryptosporidium* in the MENA region. A total of 350 faecal specimens from HIV/AIDS patients were analysed using microscopy, and a *Cryptosporidium* infection was identified from 33 samples, with 22 isolates successfully sequencing and confirming species and subtypes. Based on sequence analysis, 15 isolates were identified as *C. parvum* with family subtypes IIa (*n* = 7) and IId (*n* = 8), while five were identified as *C. hominis* (family subtypes Ia (*n* = 2) and Ib (*n* = 3)) and two as *C. felis*. The *C. parvum* subtype families IIa and IId predominated, suggesting potential zoonotic transmission. More extensive sampling of both humans and farm animals, especially sheep, goats and calves, as well as a collection of epidemiological data are needed for a better understanding of the sources of human *C. parvum* infections in Algeria.

## 1. Introduction

Diarrhoea is usually a symptom of bacterial, viral or parasitic infection and is a leading cause of death in children under five years old worldwide [1,2]. HIV/AIDS and diarrhoea are both major concerns in developing countries. The intestinal tract is one of the hardest-hit organs in HIV-infected individuals. An enterocytic–HIV infection results in enterocyte atrophy, destruction of gut immune cells and intestinal dysfunction leading to diarrhoea [3]. Diarrhoea is a significant cause of morbidity in HIV patients, and nearly 40% of those who die of AIDS experience diarrhoea [4,5,6]. People with HIV/AIDS are at increased risk of contracting diarrhoea. *Cryptosporidium* is a common cause of diarrhoea in people with HIV/AIDS [7]. This parasitic infection is an opportunistic infection indicator of full symptomatic AIDS. In people with weakened immune systems (HIV-infected people, cancer patients and organ recipients), cryptosporidiosis can be severe, long-lasting and sometimes fatal. Of the 40 currently recognised *Cryptosporidium* species, *Cryptosporidium hominis* and *Cryptosporidium parvum* are responsible for most human infections. However, other species have also been found in immunocompromised patients, including *C. meleagridis*, *C. canis* and *C. felis* [8]. Cryptosporidiosis remains a common cause of chronic diarrhoea in AIDS patients in developing countries, with up to 74% of AIDS patients with diarrhoea harbouring microorganisms in their stool [9].

The Middle East and North Africa (MENA) region encompasses approximately 22 countries that extend from Morocco to Iran. The estimated prevalence rate of adults (aged 15–49 years) living with HIV infection is one of the lowest in the world (less than 0.1%). In Morocco, although no data are available concerning the incidence of cryptosporidiosis in HIV-infected patients, two respiratory cryptosporidiosis cases were reported in this population under high active antiretroviral therapy (HAART) [10]. Algeria has a sustained low-level epidemic (prevalence in the general population in Algeria is less than 0.1%), and rates are higher among migrant and mobile populations, according to the national plan to combat sexually transmitted infections/HIV/AIDS for 2020–2024 issued by the Ministry of Health, Hospitals and Population Reform of Algeria. Available data indicate 11,000 to 14,000 people living with HIV, among whom 68–82% had access to antiretroviral therapy. However, in the country, the late detection of HIV infection and the non-adherence to treatments are individual, social and economic problems in the fight against HIV/AIDS. As with other developing countries, opportunistic intestinal parasites are poorly addressed [11]. With the development of molecular epidemiology, more and more data are available worldwide, enabling a better knowledge of the distribution of *Cryptosporidium* spp. In Algeria, even though the information is available on the distribution of *Cryptosporidium* species in animals [12,13,14], there is a lack of data on the prevalence and subtypes of *Cryptosporidium* spp. circulating in HIV/AIDS patients. This study aimed to characterise *Cryptosporidium* species and subtypes distribution in HIV-infected patients in Algeria. Not only will the findings generated from this study improve our understanding of the molecular epidemiology of cryptosporidiosis in Algeria, but it will also contribute to the mapping of the epidemiology of *Cryptosporidium* in the MENA region.

## 2. Materials and Methods

### 2.1. Patients, Faeces Sampling and Microscopy

A cross-sectional study was conducted. Faecal sampling was carried out from 2016 to 2018 and obtained from 350 patients with an HIV/AIDS positive status associated with diarrhoea attending inpatient (hospitalisation) and outpatient care units at El Hadi Flici (ex El- Kettar) hospital, Alger city, Algeria. After obtaining informed consent, patients completed a comprehensive questionnaire on age, sex, contact with animals (pets and farm animals) and drinking water sources. Clinical characteristics, including diarrhoea, weight loss, vomiting, abdominal pain and nausea, types of HAART drug regimens and laboratory characteristics, including blood CD4+ T-cell counts, were recorded by the physicians in charge. *Cryptosporidium* microscopy-based screening was performed in El Hadi Flici Ex El- Kettar hospital, Alger city, Algeria. All specimens were smeared onto glass slides, stained using the modified Ziehl Nielsen (mZN) and auramine techniques [15] and examined using light (1000×) and fluorescence (100× and 400×) microscopy, respectively (Figure 1). A sample was considered *Cryptosporidium*-positive if typical 4–6 μm diameter oocysts were visible. Positive samples were transferred to the Centre National de Référence–Laboratoire Expert crypyosporidioses (CNR-LE) (Rouen University Hospital, France) for molecular analysis.

### 2.2. Molecular Characterisation of Cryptosporidium spp.

#### 2.2.1. Identification and Characterisation of *Cryptosporidium* Species

*Cryptosporidium* species were screened using 18S rRNA gene, genomic DNAs were subjected to PCR-based sequencing as described by Khoeler et al., 2013 [16]. A two-step nested PCR protocol was used to amplify the 18S rRNA gene (215 bp). For primary PCR, the cycling protocol was 94 °C for 5 min (initial denaturation), followed by 30 cycles of 94 °C for 45 s (denaturation), 45 °C for 2 min (annealing) and 72 °C for 1.5 min (extension), with a final extension of 72 °C for 10 min. For secondary PCR, the protocol was 94 °C for 5 min, followed by 35 cycles of 94 °C for 30 s, 55 °C for 30 s and 72 °C for 30 s, with a final extension of 72 °C for 10 min. *C. hominis*, *C. parvum* and no-template PCR controls were included in each run for each protocol.

#### 2.2.2. Gp60 Sequence Amplification

Genotyping was performed by sequencing a fragment of the Gp60 gene. Primers AL3531 and AL3533 were used in the primary PCR and primers AL3532 and LX0029 in the secondary PCR leading to amplification of a fragment of approximately 364 bp [17]. Each PCR mixture (total volume, 50 μL) contained 5 μL of 10× DreamTaq Buffer, each deoxynucleoside triphosphate at a concentration of 0.2 mM, each primer at a concentration of 100 nM, 2.5 U of DreamTaq polymerase and 5µL of DNA template. Additionally, 1.25 µL of DMSO (100%) was added to the mixture. A total of 40 cycles, each consisting of 94 °C for 45 s, 55 °C for 45 s and 72 °C for 1 min, were performed. An initial hot start at 94 °C for 3 min and a final extension step at 72 °C for 7 min was also included. Each amplification run included a negative control (PCR water) and two positive controls (genomic DNA from *C. parvum* oocysts purchased from Waterborne Inc., and *C. hominis* genomic DNA from a faecal specimen collected in Rouen University Hospital). Products were visualised in 2% agarose gels using ethidium bromide staining, and identification was confirmed by sequencing. Positive samples were further genotyped by DNA sequencing of the Gp60 gene amplified by a nested PCR following the protocol described elsewhere [18].

#### 2.2.3. DNA Sequence Analysis

Sequencing was used to confirm *Cryptosporidium* species/genotypes from second-round PCR products. PCR amplicons were purified using Exonuclease I/Shrimp Alkaline Phosphatase (Exo-SAP-IT) (USB Corporation, Cleveland, OH, USA). They were sequenced in both directions using the same PCR primers at 3.2 μM in 10 μL reactions, Big Dye™ chemistries, in ABI 3500 sequence analyser (Applied Biosystems, Foster City, CA, USA). Sequence chromatograms of each strand were examined with 4 peaks software and compared with published sequences in the GenBank database using BLAST (www.ncbi.nlm.nih.gov/BLAST).

### 2.3. Consent and Ethical Approval

The authors confirm that all the participants were apprised of the aims of the study protocol. For those aged <18 years, consent was obtained from parents or guardians. Participants were also informed of the right to refuse to participate or withdraw from the study at any time without giving any reason. This study was approved by the Ethical clearance committee of the El Hadi Flici Ex El- Kettar hospital.

### 2.4. Statistical Analysis

The obtained results were presented using tables and charts (descriptive statistics). R statistical software (version 3.6.3), Chi-squared and Fisher’s exact tests were used to check for an association between *Cryptosporidium* and factors studied. *p*-values < 0.05 were considered statistically significant.

## 3. Results

### 3.1. Clinical Characteristics of Patients

Of individual faecal samples from 350 HIV patients examined for the presence of *Cryptosporidium* oocysts, 33 (16 female and 17 male patients) were found positive. The median age of these patients was 40 years (range 7–82 years). Reported cases were highest among patients aged 20–50 years (Figure 2). The major clinical symptoms consisted of watery diarrhoea in all patients (chronic in 31, intermittently in two) which may be associated with nausea, vomiting or abdominal pain (*n* = 32). Moreover, fever, asthenia and weight loss were reported in 8, 16 and 23 patients, respectively. Less frequently, headache or cognitive impairment was associated with *Cryptosporidium* infection (*n* = 5). Mean and median values of CD4+ cell counts were 81.65 cells/mm3 and 50 cells/ mm^3^ (range 1–512 cells/mm^3^), respectively. Correlation of the *Cryptosporidium* infection in HIV patients with their CD4+ cell count proved that the patients with CD4 count of <200 cells/mm^3^ are at higher risk of cryptosporidiosis, with more cases reported below 50 cells/mm^3^ CD4 count (Table 1).

### 3.2. Cryptosporidium Species and gp60 Genotypes Distribution

A total of twenty-two of the thirty-three positive isolates were successfully amplified at the 18S rRNA and gp60 locus. Based on sequence analysis, *C. parvum* was identified in 15 samples with family subtypes IIa (*n* = 7) and IId (*n* = 8). *C. hominis* was detected in 5 cases (family subtypes Ia (*n* = 2) and Ib (*n* = 3)) and 2 patients were infected with *C. felis*. Heterogeneity of *Cryptosporidium* was observed, eleven subtypes were identified, including seven *C. parvum* subtypes (IIaA14G2R1, IIaA15G2R1, IIaA16G2R1, IIaA20G1R1, IIaA21G1R1, IIdA16G1 and IIdA19G1) and four *C. hominis* subtypes (IaA24, IaA22R2, IbA10G2 and IbA13G3). Among *C. parvum* subtypes, IIdA16G1 and IIdA19G1 had the highest occurrence followed by IIaA15G2R1. For *C. hominis* isolates, IbA13G3 was identified in 2 specimens and the other subtypes in one sample each (Table 2). Unique sequences generated in this study were deposited in GenBank under accession numbers MT084775-MT084794.

## 4. Discussion

In many world regions, microscopy-based identification provides a simple, cost-effective and vital method for diagnosing and screening hematologic and infectious diseases. In this study, only microscopically positive samples for *Cryptosporidium* infection were sequenced and further analysed; however, microscopy is less sensitive, which may underestimate the prevalence of *Cryptosporidium* infection and lead to some biases in the following results. Although the low rate of positive microscopy results in our study could result from the low sensitivity of microscopy, the slides were independently confirmed by two readers, with any discrepancies resolved by a third reading. We collected one stool sample from each participant, which may result in an underestimate of the true prevalence.

Cryptosporidiosis is a significant cause of chronic diarrhoea and death in HIV/AIDS patients. Diarrhoea occurs in 90% of HIV/AIDS patients in developing countries and about 30–60% in developed countries [19,20]. According to the World Health Organization, classic diarrhoea is generally differentiated into acute and chronic-based depending on its duration. Acute diarrhoea is described as having acute onset and duration of not more than 14 days. In contrast, chronic or persistent diarrhoea is defined as having an onset of more than 14 days. Here, 31 of 33 cryptosporidiosis cases in HIV/AIDS patients are associated with chronic diarrhoea. The gastrointestinal tract is a major site of HIV replication, which results in the massive depletion of lamina propria CD4 T cells during acute infection, which leaves affected individuals mortally susceptible to opportunistic infections. In this study, six patients died from cryptosporidiosis, and two were associated with pulmonary tuberculosis and visceral leishmaniasis. Immunostimulation induced by *Leishmania* promotes HIV replication and progression to AIDS while decreasing immunity through immune resource depletion [21,22]. Access to highly active antiretroviral therapy (HAART) has significantly reduced the morbidity and mortality caused by cryptosporidiosis [23].

In the literature, it has been shown that treatment with ritonavir results in a drastic reduction in *C. parvum* infection in vivo (neonatal Balb/c mice) and in vitro (human ileal adenocarcinoma cell line) models [24]. Algeria has provided HAART free of charge since 1998, standing out as one of the countries in the MENA region with the most advanced health responses. In our present findings, the overall prevalence rate of cryptosporidiosis was 9.42% (33/350) among HIV/AIDS patients. Among the 22 patients with GP60-characterised *Cryptosporidium* spp. infection, nine documented patients reported adherence to HAART and distributed as follows: (1) Four patients initiated first-line ART regimen consisting of a combination between nucleoside analogue reverse transcriptase inhibitors (NRTIs) and non-NRTIs. Lamivudine (3TC) and Efavirenz (EFV) were commonly used as the backbone in first-line therapy. HAART regimen was diverse: 3TC/EFV/Abacavir (ABC) (*n* = 2), 3TC/EFV/ Didanosine (ddI) (*n* = 1) and 3TC/EFV/Zidovudine (AZT) (*n* = 1). The results of the Gp60 subtyping showed one *C. hominis* Ib family (IbA13G3) and within *C. parvum*, two IIa family subtypes (IIaA15G2R1 and IIaA21G1R1). (2) Four individuals were using the second-line regimen. The favoured second-line therapy was a double-boosted protease inhibitor combination regimen consisting of Darunavir (DRV) boosted with Ritonavir (RTV) in association with 3TC. Subtypes IbA10G2, IIaA15G2R1, IIaA16G2R1 and IIdA16G1 were detected. (3) *C. parvum* IIaA15G2R1 was identified in a patient with virologic failure on second-line ART regimen. Virologic failure represents the definition of viral non-suppression (plasma HIV RNA > 1000 copies/mL) used by the WHO Public health approach for low-and middle-income countries. This patient died from dehydration associated with high parasite loads. As for whether it was the first or the second-line regimen, no significant association was found between *Cryptosporidium* infection and HAART treatment at the species and subtype level. In Tunisia, a neighbouring country, 42/526 included outpatients and inpatients who exhibited *Cryptosporidium* spp. oocysts in faeces. A total of six of the forty-two positive cases were encountered in HIV/AIDS patients [25]. Higher prevalence rates were reported in other countries, such as Ethiopia, with 43.6% [26]. Even though HIV-infected patients under HAART have a reduced risk of suffering from an opportunistic infection, opportunistic gastrointestinal infection still occurs.

Currently, nitazoxanide is the only proven anti-parasitic therapy for *Cryptosporidium* infections. Although it has previously been demonstrated that nitazoxanide can improve parasite clearance and shorten the duration of clinical symptoms in adults with AIDS-related cryptosporidiosis [27], other trials could not show any significant benefit [28]. Its limited efficacy in compromised hosts has raised important questions regarding how to manage these patients. Paromomycin and/or azithromycin in conjunction with nitazoxanide have been investigated in double-blind, randomised studies to treat cryptosporidiosis in immunocompromised patients, such as those with HIV/AIDS, and the results were found to be encouraging [29]. Be that as it may, improved medications for the treatment of cryptosporidiosis are desperately needed.

*Cryptosporidium* spp. infection was reported in patients with advanced immunodeficiency who are on HAART, which may explain their dyspeptic symptoms [30]. In HIV patients, CD4+T cell counts <100 cells/mm^3^ were associated with susceptibility to the *Cryptosporidium* infection. CD4+T cell counts help predict the course of *Cryptosporidium* infection, as in many other opportunistic infections. The lower the CD4 cell count (less than 50), the more severely will the disease occur. In this study, the patients’ mean CD4+ T-cell count was 81.66 ± 98.36 cells/mm^3^ while the median was 50, which is consistent with the previous reports [31,32] stating that HIV-infected patients can have *Cryptosporidium* infection even when they are on HAART. Therapeutic intervention leads to recovery of the CD4+T cell count in HIV/AIDS patients. Established *C. parvum* infection resolution in a murine model requires CD4+Tcells and gamma interferon [33]. The emergence of drug-resistant HIV variants, failure or discontinuation of HAART, and /or the re-emergence of *Cryptosporidium* spp. infection in these patients should be seriously considered and addressed [34,35].

To our knowledge, this is the first study of *Cryptosporidium* species and subtypes distribution in HIV/AIDS patients in Algeria. *Cryptosporidium parvum* was the most common species responsible for cryptosporidiosis. The high diversity of *Cryptosporidium parvum* subtypes was observed in this study, and the results showed that infections were marked by zoonotic isolates of *C. parvum* (IIa and IId) subtypes. The most prevalent subtype in the IIa subtype family corresponds to the IIaA15G2R1 (*n* = 3/15). This subtype is the most dominant subtype infecting especially dairy cattle and has been widely reported in zoonotic infections [18,36]. As a risk factor for human cryptosporidiosis, contact with cattle or consumption of raw milk was suggested to be implicated in neighbouring countries such as Tunisia [37]. Interestingly, this subtype has never been reported in Algeria in cattle or other animals. More investigations should be performed with larger and more representative cattle samples in the country. The IId family was generally considered a sheep and goat subtype, even if it was already encountered in humans [38,39]. The IIdA16G1 subtype (*n* = 4/15) identified during this study has been recently reported in Algerian sheep [40]. The subtype IIdA19G1 (*n* = 4) was also detected and reported in goats in Spain [41]; in both HIV-positive patients and pre-weaned dairy cattle in China [9,42] but had never been reported in goats or other animals in Algeria. Analysis of the questionnaire showed that three out of eight patients harbouring IId subtypes stated (i) contact with animals or their excreta (living in rural areas of farmed livestock and as sheep breeder) and (ii) consumption of well water. A truck driver infected with *C. parvum* IIdA19G1 also noted drinking water from wells as he travelled southward. Some cryptosporidiosis outbreaks have been linked to contaminated water. Water treatment system deficiencies are frequently recognised as a primary cause of outbreaks [43,44,45]. Even the finest systems can be overwhelmed by a large density of oocysts entering the source waters in a short period. The thick-walled oocyst is resistant to environmental conditions and most disinfection methods [46]. In this study, tap water is most drunk by HIV/AIDS patients. Aragon TJ et al., in 2003, showed that tap water is the most likely source of cryptosporidiosis. Indeed, people with AIDS who drank tap water in San Francisco were 23 times more likely to contract cryptosporidiosis than those who drank bottled water [47]. Although surface waters are more vulnerable to direct contamination from sewage discharges and runoff, *Cryptosporidium* may also be found in groundwater. Hancock et al. (1998) published the research results, including 199 groundwater samples screened for *Cryptosporidium*. They discovered *Cryptosporidium* oocysts in 5% of vertical wells, 20% of springs, 50% of infiltration galleries and 45% of horizontal wells.

In Algeria, *Cryptosporidium parvum* was largely predominant over *C. hominis*, which was reported as the most frequent species in immunocompromised patients [48,49]. Sequences analysis showed the presence of IaA14, IaA22R2, IbA10G2 and IbA13G3. The IbA13G3 subtype is rarely isolated in humans but has already been reported as imported cases of cryptosporidiosis in Spain [50] and encountered in Peruvian HIV-positive individuals, as well as in Nigeria and Cameroon [51,52].

Potential zoonotic transmission to *C. felis* (*n* = 2) was highlighted during this study. *C. felis* usually affects cats; a patient infected with this species reported, in the questionnaire, close contact with cats and birds and died from neuromeningeal cryptococcosis. In Africa, reports of human infection with *C. felis* are scarce. *C. felis* was reported in HIV patients in Ethiopia [26], HIV and non-infected patients in Nigeria [31] and children under 5 years old in Kenya [53]. Anthroponotic transmission of *C. felis* can occur, particularly in areas with a high incidence of cryptosporidiosis in HIV patients [54].

## 5. Conclusions

The present study documents the occurrence of *Cryptosporidium* infection in HIV/AIDS patients in Algeria and the characterisation of *Cryptosporidium* subtypes. Not only will the findings generated from this study improve our understanding of the molecular epidemiology of cryptosporidiosis in Algeria, but they will also contribute to the mapping of the epidemiology of *Cryptosporidium* in the MENA region. Future research may be directed by the findings in this study (the predominance *of C. parvum* subtype families IIa and IId) to compare samples from humans and animals to help investigate sources and routes of transmission. More extensive sampling of both humans and farm animals, especially sheep, goats and calves, as well as a collection of epidemiological data are needed for a better understanding of the sources of *C. parvum* infections in humans in Algeria.

## Figures and Tables

**Figure 1 viruses-15-00362-f001:**
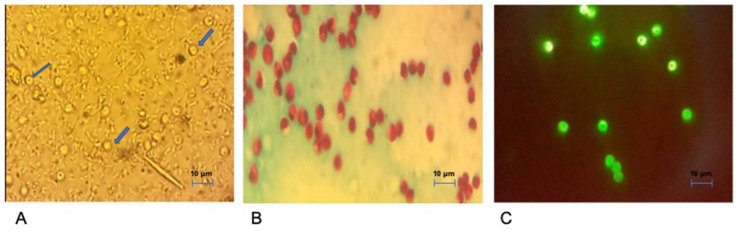
*Cryptosporidium* oocysts observed from stools (**A**) by direct wet mount (arrows); (**B**) mZN staining and (**C**) auramine–phenol staining.

**Figure 2 viruses-15-00362-f002:**
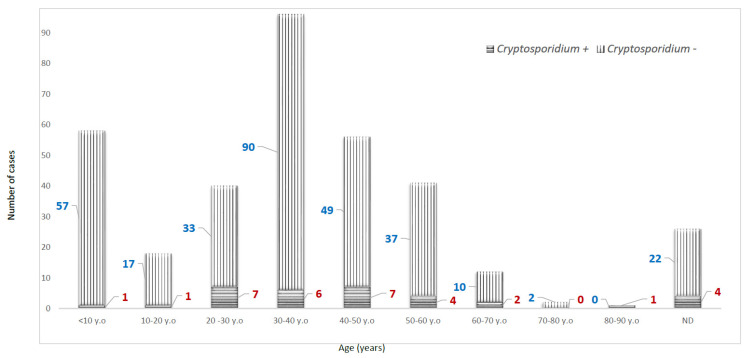
Stacked column chart depicting the age distribution of *Cryptosporidium*-infected HIV/AIDS patients. The red-coloured numbers represent the number of *Cryptosporidium*-positive patients, and the blue ones are the negatives. ND: Not disclosed.

**Table 1 viruses-15-00362-t001:** Association of CD4 count with cryptosporidiosis.

	Cryptosporidiosis			
	Positive	Negative	Total	*p*-Value
CD4 count > 200 cells/mm^3^	2	202	204	-
CD4 count [50–200] cells/mm^3^	10	85	95	<0.001
CD4 count < 50 cells/mm^3^	18	30	48	<0.001
CD4 count not available	3	-	3	
Total	33	317	350	

**Table 2 viruses-15-00362-t002:** *Cryptosporidium* subtypes distribution in HIV/AIDS patients in Algeria.

Sex	Age (years)	Contact with Animals	Water Consumption	Diarrhoea	CD4 (cells/mm^3^)	*Cryptosporidium* Species	Subtypes
F	54	Cats	Tap water	Chronic	26	*C. parvum*	IIaA15G2R1
M	64	Sheep and cattle	Well water	Chronic	45	*C. parvum*	IIdA19G1
F	30	No	Tap water	Chronic	20	*C. parvum*	IIdA19G1
M	25	No	Well water	Chronic	57	*C. felis*	
M	17	Cats and pigeons	Tap water	Chronic	No sampling	*C.felis*	
M	41	No	Well water	Chronic	1	*C. parvum*	IIdA19G1
M	48	No	Tap water	Chronic	172	*C. parvum*	IIdA16G1
F	82	No response	Tap water	Intermittent	512	*C. parvum*	IIaA21G1R1
F	NR	No	Not specified	Chronic	No sampling	*C. parvum*	IIdA19G1
M	60	No response	Tap water	Chronic	55	*C. parvum*	IIaA20G1R1
F	44	No response	Not specified	Chronic	16	*C. parvum*	IIdA16G1
F	58	No response	Tap water	Chronic	40	*C. parvum*	IIaA16G2R1
M	38	No response	Tap water	Chronic	178	*C. parvum*	IIaA15G2R1
M	41	Sheep and cattle (Sheep breeder)	Well water	Chronic	92	*C. parvum*	IIdA16G1
M	NR	No response	Not specified	Chronic	207	*C. parvum*	IIaA15G2R1
M	66	Living in rural areas of farmed livestock	Tap water	Intermittent	No sampling	*C.hominis*	IaA14
F	37	No	Bottled water	Chronic	9	*C.hominis*	IbA10G2
F	30	No	Tap water	Chronic	50	*C. parvum*	IIaA14G2R1
F	28	No	Tap water	Chronic	109	*C.hominis*	IbA13G3
F	NR	No	Tap water	Chronic	47	*C.hominis*	IbA13G3
M	7	No	Tap water	Chronic	7	*C.hominis*	IaA22R2
M	42	Sheep and cattle (Sheep breeder)	Well water	Chronic	53	*C. parvum*	IIdA16G1

NR: not reported.

## Data Availability

The original contributions presented in the study are included in the article; further inquiries can be directed to the corresponding authors.

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
