# Peer review of "Occurrence and Molecular Characterization of Cryptosporidium Infection in HIV/Aids Patients in Algeria"

_viruses, 2023, doi:10.3390/v15020362_

Round 1
Reviewer 1 Report
The manuscript by Semmani et al., focused on the occurrence and characterization of Cryptosporidium infection in HIV patients in Algeria. This is a descriptive study on the presence of Cryptosporidium strains in people living with HIV and suffering from diarrhea in Alger city area.
Comments:
1)Figure 1 is interesting but should be supplemented. While figure 1 gives us an idea about the age distribution of the patients positive for Cryptosporidium in this study, it does not tell us whether young or elderly patients are more likely to suffer from Cryptosporidium-mediated diarrhea. The ratio “patient positive for Cryptosporidium” versus “total patients” should be presented for each age group.
2)The correlation of Cryptosporidium infection in people living with HIV in regard to CD4 count is interesting but not developed enough. How did authors define the 6.36 higher risk of developing Cryptosporidium infection in people with CD4 count < 100? Can a figure similar to figure 1 be presented?
3)Some demographics of patients are missing. What is the HIV viral load for each patient? Does viral load correlate with the presence of Cryptosporidium (taking into account the 350 patients screened)?
Can authors provide more demographics such as viral load; time on ART?
4)Can authors show pictures of oocysts that they used to classify stool samples as positive for Cryptosporidium?
5)This study aims at characterizing the Cryptosporidium strains present in stools of people living with HIV in Algeria. Authors should complement their discussion with additional points on Cryptosporidium treatment. Are there any major benefits for treatment of Cryptosporidium in knowing which strain is present in people living with HIV suffering from Diarrhea? Is the strain associated to the severity of the diarrhea? Authors did discuss the presence of specific strains in relation to well versus tap water.
Minor comments
The manuscript is relatively clear and easy to understand but it suffers from minor editing issues:
1)For two words, authors should decide besides the American or British spelling (Diarrhoea (line 36) vs diarrhea; faecal (line 19) vs fecal). Either one is fine, but authors should be consistent throughout the text.
2)Line 80: I believe a period is missing after Algeria.
3)Line 142: one patient seems to be missing here. 15+17=32 not 33.
4)Spacing: were two spaces used after “:” line 158?
Are authors using one space or double-space at the start of a new sentence? (see lines 197 vs 200 or 204). Either one is fine, but text should be consistent.
Line 249: more than one space between “is” and “resistant”?
5)Line 217: In this study instead of during this study.
6)Authors alternately use no space versus one space before references throughout the manuscript. Please be consistent. I believe “viruses” require one space before each number referring to references.
Author Response
Thank you for your comments. We took them into account in our revised manuscript (Ref. No.: viruses-2084813). In this file, we have answered point by point the recommendations, suggestions and corrections needed to improve the paper.

Reviewer 2 Report
The MScript is well written but when writing "Cryptosporidum" in the text specially with-out molecular characterization , it should be written as Cryptosporidium Spp. and the parasite naming should be Italicized across the text.
The English language shall be reviewed by English language professionals, in order to make the paper in great stand.
Author Response
Thank you for your comments. We took them into account in our revised manuscript (Ref. No.: viruses-2084813).
Reviewer 3 Report
The estimated prevalence rate of adults living with HIV infection in MENA is one of the lowest in the world. To date, no data on the genetic characteristics of Cryptosporidium isolates from HIV/AIDS patients in Algeria were available. This paper aimed to identify Cryptosporidium species and subtype families prevalent in Algerian HIV-infected patients and contribute to the molecular epidemiology mapping of Cryptosporidium in the MENA region. 350 fecal specimens from HIV/AIDS patients were analyzed using microscopy, Cryptosporidium infection were identified from 33 samples, 22 isolates were successfully sequenced and confirmed species and subtypes. Based on sequence analysis, 15 isolates were identified as C. parvum with family subtypes IIa (n=7), and IId (n=8), while five were identified as C. hominis (family subtypes Ia (n=2) and Ib (n=3)) and two as C. felis. C. parvum subtype families IIa and IId predominated, suggesting potential zoonotic transmission.
Here are a few suggestions:
1. The authors should provide information about the design of the study, cross-sectional study or cohort study.
2. In this study, only microscopically positive samples for Cryptosporidium infection were sequenced and further analyzed, however, microscopy is less sensitive, which may underestimate the prevalence of Cryptosporidium infection and lead some biases of the following results.
3. The statistical charts are not standard enough. The horizontal and vertical coordinates in Figure1 are not indicated. It would be better to mark the serial number in Table 1. The clinical characteristics of the subjects should be included, and which should be consistent with the result description and discussion.
4. In Result 3.1, authors should provide detailed description and forms for the evidence of “the patient with CD4 count of <100 cells/mm3 were 6.36 times more likely to have the Cryptosporidium infections with a p-value <0.001”.
5. In Result 3.3. “Association between treatment status and Cryptosporidium infection” is not related to the main content of the article.
6. The limitation of this work should be discussed.
Author Response

(The authors gave the same response as above.)
